# Visibility Estimation Based on Weakly Supervised Learning under Discrete Label Distribution

**DOI:** 10.3390/s23239390

**Published:** 2023-11-24

**Authors:** Qing Yan, Tao Sun, Jingjing Zhang, Lina Xun

**Affiliations:** The Key Laboratory of Intelligent Computing and Signal Processing of Ministry of Education, School of Electrical Engineering and Automation, Anhui University, Hefei 230601, China; 04157@ahu.edu.cn (Q.Y.); z21301112@stu.ahu.edu.cn (T.S.); 02027@ahu.edu.cn (J.Z.)

**Keywords:** deep learning, weakly supervised learning, label distribution learning, visibility estimation

## Abstract

This paper proposes an end-to-end neural network model that fully utilizes the characteristic of uneven fog distribution to estimate visibility in fog images. Firstly, we transform the original single labels into discrete label distributions and introduce discrete label distribution learning on top of the existing classification networks to learn the difference in visibility information among different regions of an image. Then, we employ the bilinear attention pooling module to find the farthest visible region of fog in the image, which is incorporated into an attention-based branch. Finally, we conduct a cascaded fusion of the features extracted from the attention-based branch and the base branch. Extensive experimental results on a real highway dataset and a publicly available synthetic road dataset confirm the effectiveness of the proposed method, which has low annotation requirements, good robustness, and broad application space.

## 1. Introduction

Visibility refers to the maximum distance at which a target object can be seen and recognized against the sky background by a person with normal vision under current weather conditions [1]. That is, during the day, with the sky near the horizon as the background, one can clearly see the outline of a dim ground target object with a visual angle greater than 20 degrees and recognize what it is. At night, one can clearly see the luminous spots of target lights. According to the Meteorological Grade Standards for Fog and Haze [2] formulated by the China Meteorological Administration, visibility is divided into five levels according to distance: light fog, with visibility ranging from 1000 m to 10,000 m; heavy fog, with visibility ranging from 500 m to 1000 m; dense fog, with visibility ranging from 200 m to 500 m; strong dense fog, with visibility ranging from 50 m to 200 m and especially strong dense fog, with visibility less than 50 m.

Fog is a very common weather phenomenon, consisting of water vapor condensates composed of small droplets or ice crystals suspended in the atmosphere near the Earth’s surface. Haze, on the other hand, is a turbid phenomenon formed by the suspension of a large amount of smoke, dust and other particles, the causes of which are still unknown. In fog and haze, the human visual system is severely disturbed, visibility is reduced, and perception of distance becomes unreliable [3,4]. For example, in January 2021, a multi-car collision occurred in section 1026 of the Jianghuyu Expressway in Hubei Province, China, due to fog, in which about 20 cars were involved in a chain-rear-end collision, but fortunately there were no casualties. It is necessary to propose a real-time visibility monitoring system to guide people in driving in low-visibility weather such as fog and reduce accidents [5].

Existing visibility measurement methods include the use of transmission and scattering visibility meters, but these devices are expensive and difficult to deploy on a large scale. With the widespread application of deep learning, obtaining visibility values directly from a single image acquired by a camera has become a reality. Image information obtained by cameras widely deployed on highways or other roads can be used to obtain visibility values.

In recent years, some researchers have applied deep learning to visibility estimation [6]. Convolutional neural networks (CNNs) in deep learning have been widely applied in various fields, consistently demonstrating superior performance. Due to the fact that in practical applications it is often only necessary to estimate the range of visibility rather than its actual numerical value, the problem of visibility estimation is currently addressed using image classification methods. The image classification networks that have been proposed so far include AlexNet [7], ResNet [8], VGG [9], Inception [10,11,12], and so on. Shengyan Li et al. [13] and Xiuguo Zou et al. [14] directly used these classification networks for visibility estimation.

In foggy images, the distribution of fog is non-uniform, with more texture features present in the foreground of the image, while the distant regions mostly contain fog-related information, lacking actual scene texture features. To address the problem of visibility estimation more effectively, Wai Lun Lo et al. [15,16] and Jiaping Li et al. [17] choose the manual extraction of regions of interest (ROI), and Yang You et al. [18] proposed a relative CNN-RNN model, where a CNN is used to capture global information and a recurrent neural network (RNN) is used to search for the farthest area in the image. In the field of deep learning, many methods have been proposed to enhance image features and improve model robustness. Guobao Xiao et al. [19] proposed a fast two-view approximate deterministic model fitting method called LGF, which is used to handle multi-structural data disturbed by noise and outliers. This method improves the accuracy and stability of model fitting through local neighborhood preservation and global residual optimization. Recently, Guobao Xiao et al. [20] put forward a robust feature matching method based on first neighbor relation guidance (FNRG), which utilizes the neighborhood information of feature points to guide hyperplane fitting. This method demonstrates significant advantages in handling large-scale variations or rotations in data. In addition, attention mechanisms have been widely adopted, and methods such as SE-Net [21], channel attention (CA) [22] and pixel attention (PA) [23] have been proposed. These attention-based methods can significantly enhance local features in images. Tao Hu et al. [24] proposed weakly supervised attention learning for fine-grained image classification, in which the combination of bilinear attention pooling and weakly supervised localization allows for the comprehensive learning of fine-grained features in images.

Visibility levels are classified based on the actual distance range of visibility, and visibility features tend to be more similar within neighboring visibility classes, which poses challenges for feature learning. The labels corresponding to visibility images also have similar ambiguities to those in the age estimation [25] task. The distribution of fog in images is often uneven, and a single label is often difficult to describe the visibility of the image. Bin-Bin Gao et al. [26,27] pointed out the connection between label distribution learning (LDL) [28,29] and RankCNN [30], unifying RankCNN and LDL within the deep label distribution learning (DLDL) framework, and achieving optimal results in age estimation and facial attractiveness estimation.

Considering the characteristics of foggy images, non-uniformity, and fuzziness, this paper proposes a visibility estimation network based on weakly supervised attention learning and label distribution learning. Firstly, we rely on weakly supervised attention learning to extract ROI and utilize two serial branches for further feature extraction. Secondly, label distribution learning is employed to capture the fuzziness between adjacent labels of visibility features. Finally, the predictions from both branches are fused to obtain the final visibility estimation result. This approach fully leverages the non-uniform characteristics of foggy images and effectively improves the prediction accuracy.

However, deep learning requires a large amount of information from image data, and collecting these images as training data is time-consuming and labor-intensive. Currently available public datasets for visibility estimation are all synthetic datasets [31]. Collecting these data in the real world is expensive and time-consuming. In the real world, foggy weather is relatively rare, leading to the inevitable issue of class imbalance when collecting data. Additionally, visibility annotation requires professional instruments, but due to the non-uniformity of foggy weather, there may still be many erroneous annotations, necessitating secondary screening by professionals. In the face of these challenges, we propose an end-to-end method that combines weakly supervised learning with label distribution learning.

Compared to existing methods, our main contributions are as follows:(1)We propose a weakly supervised localization module to find the farthest visible area in a single image without the need for additional manpower, which is served as auxiliary information to help the network learn to capture fog features.(2)We propose a discrete label distribution learning module. The fog in foggy images is usually unevenly distributed. By using Gaussian sampling to transform the original fixed classification labels into Gaussian distribution labels, the network can learn this uneven distribution, thus better adapting to the distribution of visibility levels in the image and achieving better results.

## 2. Related Works

### 2.1. ResNet

ResNet, proposed by Kaiming He et al., enables the construction of deeper networks and faster training by introducing residual connections, and has shown excellent performance in image classification tasks. ResNet comes in various versions based on the number of convolutional layers, with ResNet18 being the one with the fewest convolutional layers, the lowest number of parameters, and the shortest training time.

The structure of ResNet18 is shown in Figure 1, which includes 17 convolutional layers and one max pooling layer, with skip connections between every two convolutional layers. The network receives input images with a resolution of 224×224×3, and after going through the convolutional and pooling layers as shown in Figure 1, feature maps $F\in R^{H\times W\times C}$ are obtained.

### 2.2. Bilinear Attention Pooling

Inspired by bilinear pooling [32], bilinear attention pooling (BAP) employs the cross-fusion of attention maps and feature maps to refine features. The introduction of attention maps allows for better extraction of local features.

As shown in Figure 2, in the feature maps $F\in R^{H\times W\times C}$, H, W, and C represent the length, width, and number of channels of the feature maps, respectively. The attention maps $A\in R^{H\times W\times M}$ are generated from F by
(1)$$A=\overset{M}{\underset{m=1}{\cup }}A_{m}=f(F)$$
where f(⋅) is a convolutional function. $A_{m}\in R^{H\times W\times 1}$ represents one of the channels of the attention maps, which indicates the network’s attention level to the feature maps F. M is the number of the channel of attention maps.

As shown in Figure 2, after obtaining the attention maps A, the attention feature maps $F_{m}$ are obtained by multiplying the attention maps A and the feature maps F element-wise according to Equation (2).
(2)$$F_{m}=A_{m}⊙F,m=1,2,\ldots ,M$$
where ⊙ denotes element-wise multiplication for two tensors.

Subsequently, global average pooling is employed to further transform $F_{m}$ into more discriminative local features $f_{m}\in R^{1\times C}$.
(3)$$f_{m}=g(F_{m})$$
where g(⋅) denotes global average pooling function.

The image features are ultimately represented as a feature vector $p\in R^{1\times MC}$, which is formed by stacking the local features $f_{m}$.
(4)$$p=\Gamma (A,F)=\left(\begin{array}{c} g(A_{1}⊙F)\\ g(A_{2}⊙F)\\ \ldots \\ g(A_{M}⊙F) \end{array}\right)=\left(\begin{array}{c} f_{1}\\ f_{2}\\ \ldots \\ f_{M} \end{array}\right)$$

### 2.3. Attention-Guided Location

During the process of BAP, the attention maps $A\in R^{H\times W\times M}$ are generated. First, for a single image, the attention maps $A\in R^{H\times W\times M}$ can be utilized to generate a mask that localizes the most important region in the image. The attention map $A_{m}\in R^{H\times W\times 1}$ is determined by random selection from the attention maps $A\in R^{H\times W\times M}$. Subsequently, the attention feature map $A_{m}$ is normalized to obtain an attention normalization map $A_{m}^{*}$ by
(5)$$A_{m}^{*}=\frac{A_{m}-\min(A_{m})}{\max(A_{m})-\min(A_{m})}$$

Then, by utilizing the attention normalization map $A_{m}^{*}$, a mask $C_{m}$ is generated to obtain the bounding box $B_{m}$. By setting a threshold $\theta_{c}\in [0.4,0.6]$, $A_{m}^{*}(i,j)$ that is larger than $\theta_{c}$ is set to 1, while if it is smaller than $\theta_{c}$, it is set to 0.
(6)$$C_{m}(i,j)=\left\{\begin{array}{l} 1,\;\mathrm{if}\;A_{m}^{*}(i,j)>\theta_{c}\\ 0,\;\mathrm{otherwise} \end{array}\right.$$

Furthermore, the mask $C_{m}$ is utilized to generate the bounding box $B_{m}$ that covers the furthest visible area. Finally, the region within the bounding box $B_{m}$ is extracted from the original image and resized to the same resolution as the original image for subsequent input. The attention-guided location allows for the elimination of irrelevant features and facilitates the extraction of fine-grained features.

### 2.4. Discrete Label Distribution Learning

For the task of visibility estimation, images captured by cameras often encompass a wide range. Due to the non-uniformity of fog distribution, different sub-regions of the same image may correspond to different visibility values. The original single label fails to effectively represent the visibility features of the image. Due to the inherent ambiguity, we introduced discrete label distribution learning, which converts the traditional single label into a set of label vectors that describe the visibility distribution of the entire image.

In deep learning classification methods, usually a label corresponds to an image sample, and for a dataset $X=\{X_{1},X_{2},\ldots ,X_{N}\}$, where N represents the total number of samples, there are usually corresponding class labels $Y=\{y_{1},y_{2},\ldots ,y_{N}\}$. A single label $y_{i}$ can be converted into its corresponding label distribution $d_{i}=\{d_{1}^{i},d_{2}^{i},\ldots ,d_{K}^{i}\}$, where K represents the total number of categories. $D=\{d_{1},d_{2},\ldots ,d_{N}\}$ constitutes the novel labels for the dataset $X=\{X_{1},X_{2},\ldots ,X_{N}\}$.

To generate label distribution $d_{i}=\{d_{1}^{i},d_{2}^{i},\ldots ,d_{K}^{i}\}$ by Gaussian probability distribution, the formula for Gaussian distribution is
(7)$$f(y)=\frac{1}{\sqrt{2\pi }\sigma }\exp\left(-\frac{{\left(y-\bar{y}\right)}^{2}}{2\sigma^{2}}\right)$$
where $\bar{y}$ is the mean of the Gaussian distribution, and σ is the standard deviation of the Gaussian distribution.

According to Equation (7), a label $y_{i}$ can be converted into a label distribution $d_{i}$, in which there are two unknown variables $\bar{y}$ and σ. The $\bar{y}$ is determined by the original label $y_{i}$, and the σ is determined by experiments, which will be analyzed in Section 4.4.1. To generate a label distribution $d_{i}$ by
(8)$$d_{k}^{i}=f(k),k=1,2,\ldots ,K$$
where $d_{k}^{i}$ represents the probability of an image being in label k.

The label distribution $d_{i}$ is obtained by direct sampling does not satisfy the requirement of a probability distribution, as $\sum_{k}d_{k}^{i}=1$. To ensure that $\sum_{k}d_{k}^{i}=1$, the Euclidean norm needs to be calculated by
(9)$$\left\|d_{i}\right\|=\sqrt{\sum_{k}d_{k}^{i}{}^{2}}$$

Dividing each component by the Euclidean norm gives the final labelling distribution $d_{i}$.
(10)$$d_{i}=\frac{\left\{d_{1}^{i},d_{2}^{i},\ldots ,d_{K}^{i}\right\}}{\left\|d_{i}\right\|}$$

## 3. Methods

The proposed method is a visibility classification model that integrates weakly supervised localization. The attention mechanism is employed to generate the farthest visible region in the original fog image, where the red box in Figure 3b represents the region located by the proposed network. Compared with previous transfer learning methods [33,34], this approach effectively retains only the information in the image that is more favorable for visibility estimation, thus effectively reducing many irrelevant regions in the original image. In contrast to some deep learning methods that focus on data augmentation [35,36,37], this method employs an end-to-end deep learning model that does not require manual annotation of the farthest visible region in the original image. As a result, the difficulty of obtaining training data is greatly reduced, providing the feasibility of building a wider range of large-scale datasets for future research.

### 3.1. Overview

The method proposed in this paper consists of two branches, namely the base branch (BB) and the attention-based branch (AB), as shown in Figure 4. First, we input the processed foggy images into the neural network after uniformly resizing the images to a resolution of 224×224×3. Then, the input data is fed into the base branch, where the Bilinear Attention Pooling Module (BAPM) in the base branch generates a mask to process the foggy image of the original input, before it is fed into the attention-based branch. The attention-based branch has a similar architectural structure to the base branch in that both branches produce neural network output logits. Additionally, it should be noted that the dimensions of the logits produced by the two branches are identical. Finally, the network fuses the logits to obtain the predicted head.

### 3.2. Base Branch

As shown in Figure 4, the base branch comprises three main components. The first component is the feature extractor of the model. The feature extractor can be any neural network model. In this work, considering the training and inference efficiency, we adopted ResNet18, as shown in Figure 1. The second component is the BAPM based attention mechanism, which generates feature vectors and localization masks through attention maps. The third component is the discrete label distribution learning module (DLDLM), which converts the original label into the label distribution and optimizes the network with KL divergence loss.

### 3.3. Attention-Based Branch

As shown in Figure 4, the attention-based branch receives the attention maps generated by the base branch and ultimately produces feature vectors. In this paper, the attention-based branch has a structure similar to the base branch, in which the two feature extractors have the same structures and share weights. With the attention feature maps, we can enhance the image information more efficiently. As described in Section 2.3, we employ the mask to generate the bounding box that covers the farthest visible area. Subsequently, we extract the region inside the bounding box from the original image and resize it to the same resolution as the original image as the input to the attention-based branch. With the base branch and the attention-based branch, we obtain two feature vectors of the same dimension. By summing and averaging the feature vectors, the predicted head is obtained.

### 3.4. Loss Function

As previously mentioned, our method comprises two branches, the base branch and the attention-based branch, which generate two logits under the supervision of the original label and the label distribution, respectively. Regarding the two different outputs, two distinct loss functions are utilized.

Since the output of the neural network might be less than zero, this may result in an issue when calculating the KL divergence loss and cross-entropy loss in the next step. To solve this problem, we adopted the SoftMax function to transform the original output to a probability distribution with a range of [0, 1].
(11)$$p(d_{k}^{i}\left|X_{i}\right.)=\frac{e^{h(X_{i},d_{k}^{i})}}{\sum_{k=1}^{K}e^{h(X_{i},d_{k}^{i})}}$$
where $h(X_{i},d_{k}^{i})$ denotes the output of the neural network.

In the base branch, as the label $y_{i}$ of a single image has been transformed into a label distribution $d_{i}$, we cannot use the cross-entropy loss commonly used in classic classification models. Therefore, we choose the KL divergence loss described in Equation (12) to calculate the model’s loss, which is used for backpropagation to optimize the network parameters.
(12)$$Loss_{KL}=\frac{1}{N}\sum_{i=1}^{N}\sum_{k=1}^{K}d_{k}^{i}\ln\frac{d_{k}^{i}}{p(d_{k}^{i}\left|X_{i}\right.)}$$

In the attention-based branch, we utilize the cross-entropy loss function.
(13)$$Loss_{CE}=-\frac{1}{N}\sum_{i=1}^{N}y_{i}^{}\log p(y_{i}^{}\left|X_{i}\right.)$$

Based on the distinct loss functions in the two branches, the overall loss function of this method is defined by
(14)$$Loss=Loss_{CE}+\lambda *Loss_{KL}$$
where λ denotes a hyper parameter to balance $Loss_{CE}$ and $Loss_{KL}$.

## 4. Experiment

In order to validate the effectiveness of the method, we conducted extensive experiments on two completely different datasets. These datasets were the synthetic foggy road image database (FRIDA) [38,39,40] and the real foggy image dataset (RFID). In addition, we also compared our results with some deep learning-based multi-classification models such as AlexNet, VGG16, and ResNet, as well as fog image enhancement models such as VisNet [41], SCNN [42] and TVRNet [43].

### 4.1. Dataset

#### 4.1.1. RFID

This dataset was constructed with the assistance of the Anhui Meteorological Bureau and consists of real fog images captured by 84 surveillance cameras distributed on highways in Anhui Province, with a resolution of 640×480. The visibility labels of each image were measured by visibility observers in meteorological stations within 1 km of the surveillance cameras. Due to the non-uniform and constantly changing nature of foggy weather and the distance error between the surveillance cameras and the visibility observers, many images still suffer from labelling errors. With the assistance of meteorological experts from the bureau, we finally screened out 5149 images to construct the entire dataset. According to the fog warning level and practical needs, we divided the images into 6 categories according to visibility: 0–50 m, 50–100 m, 100–200 m, 200–500 m, 500–1000 m, and more than 1000 m. However, foggy weather is relatively rare in daily life and the collected image data are also unevenly distributed, with fewer images in categories with lower visibility. In the actual training, 150 images were selected from each category as a test set and the remaining images were used as a training set.

#### 4.1.2. FRIDA

FRIDA is a synthetic fog image dataset consisting of 84 different road scenes and 4 different fog conditions. All images in this dataset were generated through the same algorithm, which combines different road scenes, fog conditions, and visibility levels. The dataset is divided into nine visibility levels, with level 9 having the lowest visibility and level 1 having the highest visibility. The entire dataset contains 3024 images with a resolution of 640×480. In actual training, 70% of each category was selected as the training set and the remaining 30% as the test set. The number of images in each dataset is shown in Table 1, and some image data from the dataset is shown in Figure 5.

### 4.2. Evaluation Index

In order to compare with other excellent methods, this paper uses classification accuracy (ACC), root mean square error (RMSE), and F1-score as evaluation metrics.

ACC represents the percentage of correctly classified visibility level samples in the total number of samples and is defined by
(15)$$ACC=\sum_{i=1}^{N}\left[y_{i}={y^{\prime }}_{i}\right]/N$$
where [⋅] denotes the truth value operator, N is the total number of image samples, and $y_{i}$ and ${y^{\prime }}_{i}$ are the ground truth and predicted values for the sample $X_{i}$.

The RMSE represents the magnitude of the error between the predicted values and the ground truth. It is defined by
(16)$$RMSE=\sqrt{\frac{1}{N}\sum_{i=1}^{N}{\left(y_{i}-y_{i}{}^{\prime }\right)}^{2}}$$

The F1-score is a harmonic mean of precision and recall, which measures the overall level of a model’s classification. In binary classification, it is defined by
(17)$$F1=\frac{2*Precision*Recall}{Precision+Recall}$$
where precision and recall are defined by
(18)$$Precision=\frac{TP}{TP+FP},\;Recall=\frac{TP}{TP+FN}$$
where TP (true positive) refers to the number of samples that are actually positive and classified as positive by the classifier, FP (false positive) refers to the number of samples that are actually negative but classified as positive by the classifier, and FN (false negative) refers to the number of samples that are actually positive but classified as negative by the classifier.

In multi-class classification, this paper uses macro-F1 to calculate the F1-score, which refers to the average of F1-scores for each class and is defined by
(19)$$\mathrm{macro-F}1=\frac{1}{K}\sum_{k=1}^{K}F1_{k}$$

### 4.3. Implementation Details

Both the method in this paper and the comparison methods are implemented using Pytorch. The batch size is 48 and the learning rate is 0.001 decaying by 0.9 times every 5 rounds. All models are trained using the Adam optimizer for 100 epochs. The remaining details are shown in Table 2.

### 4.4. Experimental Results and Discussions

#### 4.4.1. Hyper-Parameter Settings

The method proposed in this paper includes 4 hyper-parameters: the number of attention maps channels M for bilinear attention pooling, the threshold $\theta_{c}$ of mask $C_{m}$, the standard deviation σ for discrete label distribution learning, and the weight λ for the loss function. To find the most suitable parameters, this paper analyzes the impact of each parameter on the overall model, finds a reasonable range for each parameter, and then finds the optimal combination.

Firstly, this paper used a model without a discrete label distribution learning module to study the impact of the number of attention feature map channels on overall model performance. As shown in Table 3, different values of M will have different effects on the classification results. When M is adjusted to an appropriate range, it can significantly improve the model’s classification accuracy. If M is too small, the network may fail to learn the more important features. On the contrary, if M is too large, the network may pay too much attention to specific features, leading to overfitting, which is not favorable for future classification. For two completely different datasets, the setting of M should not be the same due to the different distributions of fog features in the synthetic and real datasets. Therefore, in the subsequent experiments, we set M=2 for the RFID dataset and set M=16 for the FRIDA dataset separately.

Secondly, this paper added a discrete label distribution learning module to the above experiments, aiming to investigate the impact of varying standard deviations σ on overall model performance. As shown in Figure 6, these two images illustrate the label distributions corresponding to two different original labels. The label distribution refers to the probability of each visibility level in the image. Different curves in the same graph represent different label distributions due to different standard deviations. A larger standard deviation σ indicates higher probabilities for each visibility level in the image, indicating an uneven distribution of fog in the image. Conversely, a smaller standard deviation σ indicates that the visibility probabilities are concentrated at the same level, suggesting a uniform fog distribution in the image. The changes in visibility probability brought by different standard deviations σ can be clearly observed in Figure 6.

Table 4 demonstrates the influence of different standard deviations σ on the performance of the model in learning from the discrete label distribution. It is shown that in the visibility estimation task of this paper, a smaller standard deviation σ is significantly superior to a larger σ. This suggests that in the two datasets used in this paper, as there are 6 and 9 levels, respectively, the non-uniform fog features mainly shift to the adjacent visibility ranges of the original labels. A large standard deviation σ causes the discrete label distribution to be too scattered, affecting the model’s search for features in the farthest region. Therefore, the standard deviation σ should be adjusted to enable the model to search for features in the farthest region more effectively. In the subsequent experiments, we set σ=0.2 for both datasets.

Thirdly, this paper introduced an attention-based branch to enhance the original model’s extraction of visibility features, exploring the impact of the two branches on overall model performance by adjusting the weight λ between the loss functions of the base branch and the attention-based branch. Table 5 shows the effect of the variation of the weight λ on the experimental results in learning from the discrete label distribution. A larger λ indicates a greater weight of the base branch in optimizing the model. As shown in Table 5, the performance of the model reaches the optimal value when λ is in an appropriate range, at which time the fusion of the two different loss functions in the two branches is more optimal and can enhance the classification accuracy of the model more effectively. A larger or smaller value of λ would have a negative impact on the model. Therefore, in the subsequent experiments, we set λ=6 for the RFID dataset and set λ=4 for the FRIDA dataset, respectively.

Subsequently, the following experiment continued to study the impact of the mask $C_{m}$ threshold $\theta_{c}$ on overall performance. Table 6 shows the influence of different $\theta_{c}$ on the performance of the attention-based branch. The results indicate that an appropriate $\theta_{c}$ can better improve the performance of the model. In the two datasets used in this paper, $\theta_{c}$ has a greater impact on model performance on the synthetic dataset FRIDA, where a larger $\theta_{c}$ causes the weakly supervised localization module to select a smaller region, providing the model with incorrect local features and causing a significant decrease in accuracy. A smaller $\theta_{c}$ causes the localization region to be closer to the original image without focusing on local features, reducing the model’s accuracy. Therefore, $\theta_{c}$ should be adjusted to enable the model to locate the appropriate size of the farthest region. Therefore, in the subsequent experiments, we set $\theta_{c}$ to a randomly selected value from the range [0.4,0.6] for the RFID and FRIDA dataset to improve the robustness of the model.

Finally, the best combinations were selected based on the hyperparameter experiments, with M=2, σ=0.2, λ=6, $\theta_{c}\in [0.4,0.6]$ for the RFID dataset and M=16, σ=0.2, λ=4, $\theta_{c}\in [0.4,0.6]$ for the FRIDA dataset. These combinations were then compared to other state-of-the-art visibility estimation methods.

#### 4.4.2. Comparison with State-of-the-Art Methods

In order to demonstrate the effectiveness of our proposed method, we compared it with other classic classification methods and some improved methods based on visibility classification, as shown in Table 7. First, we conducted transfer learning experiments on two fog image datasets, RFID and FRIDA, using several classic deep learning classification models, such as AlexNet, VGG16, ResNet18, and ResNet50. Our proposed method achieved the best performance in terms of accuracy, mean squared error, and F1-score. It is worth noting that our method is improved based on the ResNet18 network, which indicates that the combination of the introduced BAP, weakly supervised localization module, and discrete label distribution learning helps to enhance the local feature information in the image. Secondly, we compared our method with several models proposed for the visibility estimation task, such as SCNN, TVRNet, and VisNet. As shown in Table 7, our method significantly outperforms these methods. This indicates that our method is more beneficial for enhancing fog features than data augmentation relying on image preprocessing in VisNet.

Figure 7 illustrates the accuracy curves during the training and testing stages of the proposed method and four baseline methods including ResNet18, VGG16, VisNet, and TVRNet. It is obvious from the figure that the method proposed in this paper has faster and more stable convergence in the training stage, and the accuracy curves in the testing stage are also comparable. Therefore, it can be concluded that the proposed method outperforms other baseline methods in terms of model stability and prediction accuracy.

Figure 8 demonstrates the classification result achieved by the proposed method and the corresponding farthest visible region of fog. The red box in the figure comes from the weakly supervised localization module, which is guided by the attention map. It can clearly be seen that the image area that the attention focuses on is concentrated in the farther visible region. When visibility is high, attention tends to focus on the farther regions of the road, while in low visibility conditions, it pays more attention to the nearby regions containing road texture details.

#### 4.4.3. Ablation Study

The innovations of our proposed method mainly include two aspects: the introduction of dual-branch weakly supervised localization based on bilinear attention pooling and discrete label distribution learning. To verify their effectiveness, we conducted ablation experiments on the combination of these two modules, and the experimental results are shown in Table 8.

First, we investigated the impact on the method of introducing the dual-branch weakly supervised localization based on bilinear attention pooling. Specifically, we considered three scenarios: ResNet18, ResNet18 combined with the base branch, and ResNet18 combined with both the base and attention-based branch. The experimental results show that after adding the BAP, the model can utilize the attention mechanism to focus on local features in the image. After adding the attention-based branch, the image from the weakly supervised localization module can enhance the feature region information that is strengthened by attention in the base branch, thereby improving the accuracy of the model.

Furthermore, we further investigated the impact of incorporating the discrete label distribution learning module into both the base branch and the attention-based branch. The results show that adding the discrete label distribution only to the base branch has a positive impact on the network, while adding it to the attention-based branch has a negative impact on the model. This suggests that the visibility feature distribution in the farthest fog-visible region generated in the base branch is more concentrated, and the global scattered distribution of visibility features no longer needs to be described by a discrete label distribution. Therefore, incorporating only the discrete label distribution into the base branch is more advantageous for the visibility estimation task in our proposed method.

#### 4.4.4. Analysis of Various Visibility Levels

The predicted results for different visibility levels on the RFID and FRIDA datasets are shown in Table 9 and Table 10, respectively. It is worth mentioning that the labels of the FRIDA do not contain actual visibility values. Level 1 corresponds to the highest visibility, while level 9 corresponds to the lowest visibility, which is in contrast to the RFID. It can be seen that the classification accuracy of all methods is relatively low in the categories with low visibility levels. On the one hand, low visibility is rarely encountered in real life, and the relatively small amount of low visibility data in our dataset leads to fewer features that can be learnt by the methods, which results in lower classification results. On the other hand, in low visibility, the method can extract fewer useful features due to the severe atmospheric scattering and the limited structural features retained in the image.

Compared with other methods, the method proposed in this paper has significantly improved accuracy in the intermediate visibility levels. As shown in Table 9, the proposed method has significant advantages over other similar methods in categories 2–5. Its success lies in the addition of the learning of the discrete label distribution, which can better represent the features with non-uniform fog concentration in the intermediate level and remove the features that may be misclassified as other levels from the base branch, thus obtaining better classification performance. The results in Table 10 show the significant improvement of our proposed method in all categories of visibility levels, especially in the low visibility levels 6–9, where no other methods can achieve good performance. Our proposed method effectively analyzes the non-uniform phenomenon of fog concentration in the image and more accurately extracts the features belonging to the corresponding visibility levels, thus improving the accuracy of estimating the visibility level of fog images. Figure 9 displays the confusion matrix of the proposed method on two datasets. It can be observed that the proposed method predicts adjacent labels in the event of prediction failures, which is acceptable in real-life scenarios.

#### 4.4.5. Algorithm Complexity Study

We also studied the computational resource consumption of several deep learning methods. Table 11 shows the training and validation time, spatial and temporal complexity, and number of parameters required by various methods on the RFID dataset. They use the same hardware and software environment, and the image resolution input to the network is standardized to 224×224×3. In deep learning methods, the main consumption of computational resources comes from the parameter calculation process during feature extraction and backpropagation, and time complexity mainly affects the training and validation time of the model. The increase in training time is due to the weakly supervised localization module based on BAP introduced in this paper. When the number of channels M in the attention map is increased, the amount of parameters to be calculated during training will increase to a certain extent. However, in the hyperparameter experiments in Section 4.4.1, we found that M was not necessarily better as it became larger. We chose a smaller M under the same accuracy rate to maintain a balance between performance and efficiency. In addition, the experimental results show that the other hyperparameters have no impact on the overall efficiency of the model. Compared with the base model ResNet18, our proposed method has only a slight increase in training time, and the required time for validation is roughly the same or even shorter. Compared with other deep learning methods, our proposed method has the shortest validation time for the real-time detection of visibility levels. The experimental results show that the increase in computational resource consumption is acceptable compared to the improved classification performance.

## 5. Conclusions

This paper proposes a novel end-to-end deep learning method for visibility estimation of foggy images. Due to the non-uniform distribution of fog concentration in foggy images, it is difficult for a single category label corresponding to a single image to be advantageous for network learning. Therefore, it is essential to fully exploit the useful information in non-uniform foggy images. This method employs the method of discrete label distribution to optimize the extraction of non-uniform fog concentration information in foggy images, along with the bilinear attention pooling method in weakly supervised localization to extract the farthest visible fog regions in images that favor visibility information extraction, enabling the network to fully exploit the local features in the image. In real weather conditions, images with low visibility are rare. In order to balance the data volume of different categories in the dataset, the overall size of the dataset can only be maintained at a low level. However, data-driven deep learning models require a large amount of image data. Therefore, this paper spent a lot of time building a dataset containing 5149 real images. To evaluate the performance of this method, we conducted experiments on two datasets, the RFID and FRIDA, and compared our method with other state-of-the-art deep learning-based methods. The results of extensive experiments indicate that the method proposed in this paper is more effective and needs less annotated information in the training set compared to other methods. In addition, by using an efficient ResNet18 as the backbone and weight sharing, the overall network’s performance requirements were maintained at a low level. The weakly supervised localization module based on BAP and the DLDLM has a small number of additional parameters, making it better able to meet real-time requirements in road visibility warning and traffic control.

## Figures and Tables

**Figure 1 sensors-23-09390-f001:**
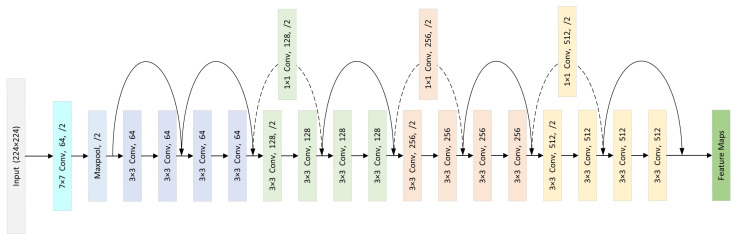
The overall structure of ResNet18.

**Figure 2 sensors-23-09390-f002:**
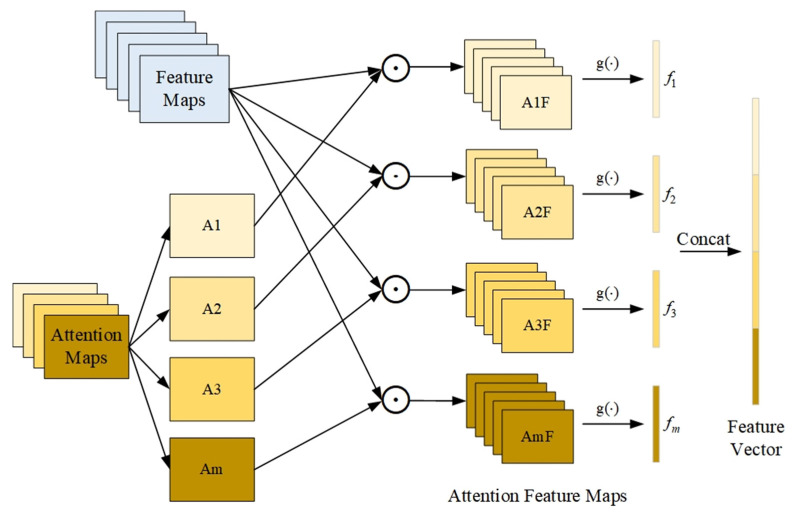
The process of bilinear attention pooling.

**Figure 3 sensors-23-09390-f003:**
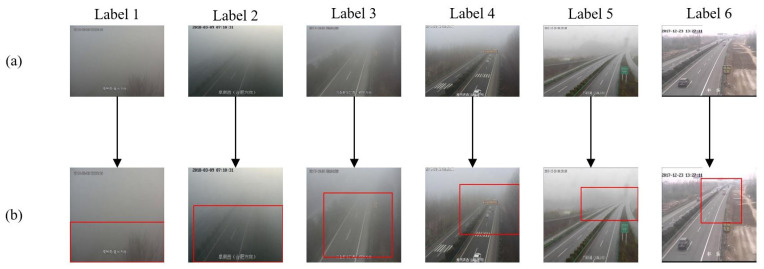
The network detects the farthest region from the image. In (**a**), the original fog images of each visibility level are shown. In (**b**), the red box indicates the farthest visible area extracted by our method.

**Figure 4 sensors-23-09390-f004:**
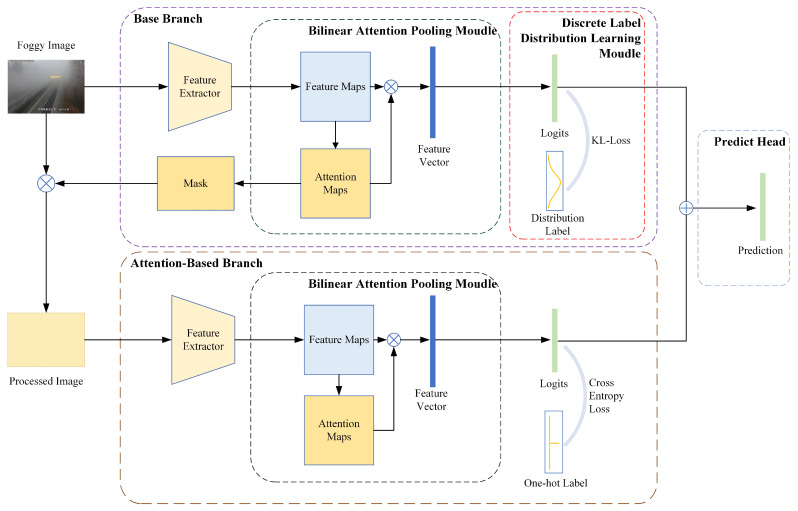
Overview of the process of our method.

**Figure 5 sensors-23-09390-f005:**
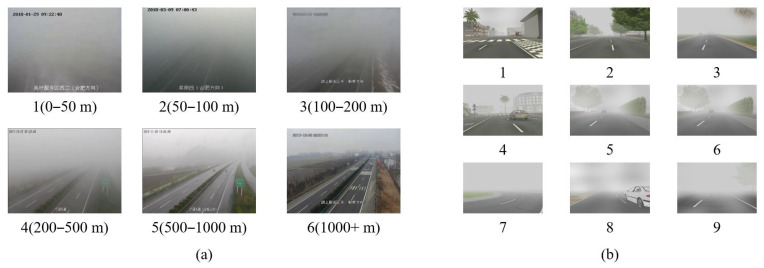
Images from the RFID with 6 visibility levels (**a**), from the FRIDA with 9 visibility levels (**b**).

**Figure 6 sensors-23-09390-f006:**
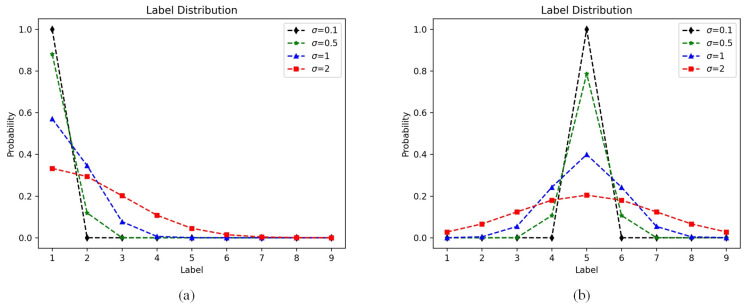
The different discrete label distribution with different σ in FRIDA. (**a**) is the discrete label distribution corresponding to the original label 1 (the highest level of visibility). (**b**) is the discrete label distribution corresponding to the original label 5.

**Figure 7 sensors-23-09390-f007:**
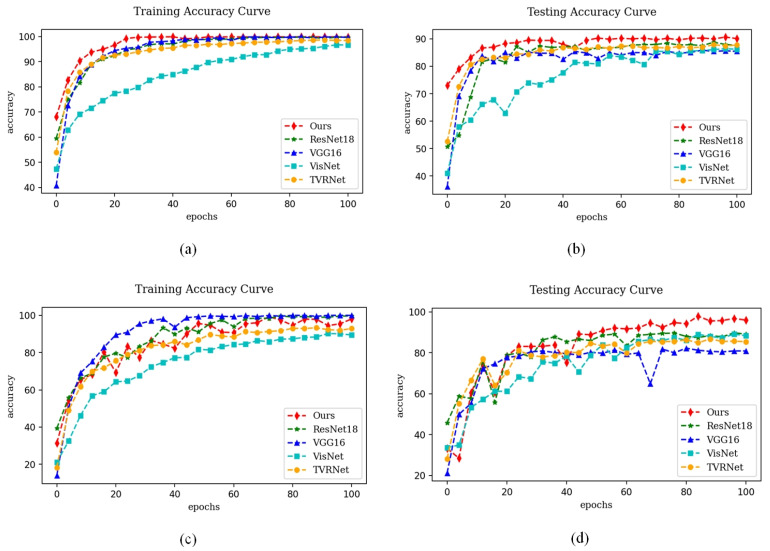
Accuracy curves on two datasets. (**a**,**b**) shows the training and testing accuracy curve with the RFID; (**c**,**d**) shows the training and testing accuracy curve with the FRIDA.

**Figure 8 sensors-23-09390-f008:**
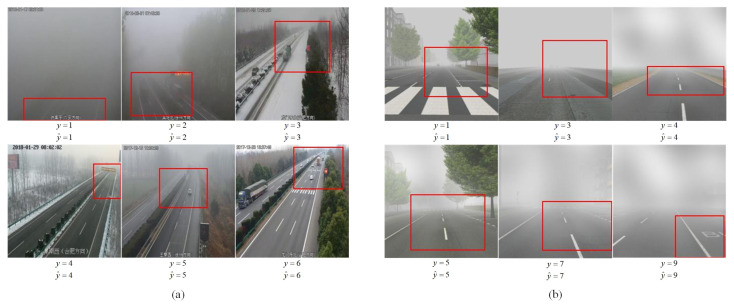
Our classification result ($\hat{y}$) and the ground truth (y). The red frames are the farthest visible regions of fog detected by the proposed method. (**a**) The result of the RFID; (**b**) The result of the FRIDA.

**Figure 9 sensors-23-09390-f009:**
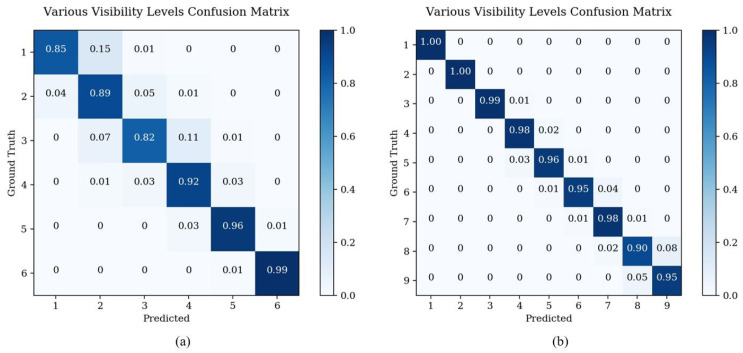
Confusion matrices of the proposed method. (**a**) Confusion matrix of the RFID; (**b**) confusion matrix of the FRIDA. Darker colors in the figure represent higher accuracy.

**Table 1 sensors-23-09390-t001:** The distribution of our datasets with detailed levels and quantities.

Datasets	1	2	3	4	5	6	7	8	9	Total
RFID	304	622	774	891	948	1610	/	/	/	5149
FRIDA	336	336	336	336	336	336	336	336	336	3024

**Table 2 sensors-23-09390-t002:** Experimental environment.

Item	Content
CPU	Intel Xeon(R) Silver 4214R
GPU	RTX 3080Ti
RAM	128 GB
Operating System	Ubuntu 20.04.4 LTS
Programming Language	Python 3.7.9
Deep Learning Framework	Pytorch 1.7.1
CUDA Version	11.0

**Table 3 sensors-23-09390-t003:** The mean ACC, RMSE, and F1-score of our network under different values of hyper-parameter M with the RFID and FRIDA dataset. Bold numbers in the table represent the best performances.

M	RFID			FRIDA		
	Accuracy (%) ↑	RMSE ↓	F1 ↑	Accuracy (%) ↑	RMSE ↓	F1 ↑
1	89.67	0.3677	0.8985	92.89	0.277	0.9288
2	**90.22**	0.3592	**0.9047**	93.11	0.2541	0.9312
4	90.11	0.3393	0.9020	94.78	0.2252	0.9479
8	90.00	**0.3382**	0.9042	95.67	**0.2016**	0.9565
16	89.78	0.3697	0.8985	**96.00**	0.2024	**0.9599**
32	89.67	0.3566	0.8972	95.11	0.2297	0.9511
64	89.67	0.3404	0.8972	95.00	0.2165	0.9500

**Table 4 sensors-23-09390-t004:** The mean ACC, RMSE and F1-score of our network under different hyper parameter values with the RFID and FRIDA dataset. Bold numbers in the table represent the best performances.

σ	RFID			FRIDA		
	Accuracy (%) ↑	RMSE ↓	F1 ↑	Accuracy (%) ↑	RMSE ↓	F1 ↑
0.1	90.00	0.3409	0.8969	95.22	0.2197	0.9518
0.2	**90.22**	**0.3409**	**0.9026**	**96.11**	**0.1954**	**0.9611**
0.5	89.67	0.3647	0.8962	94.22	0.2327	0.9421
1	88.78	0.3983	0.8885	94.22	0.2364	0.9425
1.5	89.00	0.3923	0.8729	90.00	0.3146	0.8987
2	88.00	0.4069	0.8712	89.33	0.3244	0.8892

**Table 5 sensors-23-09390-t005:** The mean ACC, RMSE and F1-score of our network under different hyper parameter values with the RFID and FRIDA dataset. Bold numbers in the table represent the best performances.

λ	RFID			FRIDA		
	Accuracy (%) ↑	RMSE ↓	F1 ↑	Accuracy (%) ↑	RMSE ↓	F1 ↑
1	90.22	**0.3409**	0.9026	96.11	0.1954	0.9611
2	89.67	0.3764	0.8900	96.00	0.2067	0.9599
4	90.11	0.3613	0.9030	**96.78**	**0.1882**	**0.9677**
6	**90.67**	0.3456	**0.9058**	94.78	0.2213	0.9477
8	90.22	0.3584	0.9026	94.67	0.2444	0.9466
10	89.89	0.3589	0.8988	95.11	0.222	0.9513
15	89.44	0.3814	0.8935	95.67	0.2016	0.9567
20	89.89	0.3516	0.8994	95.44	0.2228	0.9544

**Table 6 sensors-23-09390-t006:** The mean ACC, RMSE and F1-score of our network under different values for hyper-parameter $\theta_{c}$ with the RFID and FRIDA dataset. Bold numbers in the table represent the best performances.

$\theta_{c}$	RFID			FRIDA		
	Accuracy (%) ↑	RMSE ↓	F1 ↑	Accuracy (%) ↑	RMSE ↓	F1 ↑
0.1	89.56	0.3478	0.8960	94.67	0.2302	0.9466
0.2	89.44	0.3566	0.8947	95.67	0.2086	0.9567
0.3	90.11	0.3452	0.9014	96.00	0.1987	0.9600
0.4	90.22	0.3424	0.9002	96.44	0.1893	0.9644
0.5	90.44	0.3409	0.9020	**96.78**	**0.1882**	**0.9677**
0.6	**90.67**	**0.3377**	**0.9024**	96.33	0.1921	0.9633
0.7	89.78	0.3489	0.8984	94.22	0.2418	0.9421
0.8	89.11	0.3662	0.8907	90.22	0.3170	0.9014
0.9	88.44	0.3785	0.8851	73.89	0.5861	0.7057

**Table 7 sensors-23-09390-t007:** Comparison with state-of-the-art methods on the RFID and FRIDA datasets. Bold numbers in the table represent the best performances.

Method	RFID			FRIDA		
	Accuracy (%) ↑	RMSE ↓	F1 ↑	Accuracy (%) ↑	RMSE ↓	F1 ↑
AlexNet	85.78	0.4706	0.7641	85.00	0.4971	0.7484
VGG16	85.89	0.4360	0.8140	82.22	0.5538	0.7305
ResNet18	88.78	0.4074	0.8125	89.78	0.3284	0.7991
ResNet50	89.44	0.3722	0.8273	89.33	0.3398	0.7645
SCNN	88.33	0.3819	0.8837	88.44	0.3521	0.8838
TVRNet	87.89	0.4430	0.8790	86.78	0.3814	0.8654
VisNet	86.44	0.4295	0.8652	89.11	0.3300	0.8918
Ours	**90.67**	**0.3456**	**0.9058**	**96.78**	**0.1882**	**0.9677**

**Table 8 sensors-23-09390-t008:** Contribution of proposed components and their combinations.

Method	RFID			FRIDA		
	Accuracy (%) ↑	RMSE ↓	F1 ↑	Accuracy (%) ↑	RMSE ↓	F1 ↑
ResNet	88.78	0.4074	0.8125	89.78	0.3284	0.7991
ResNet + BB	89.44	0.3550	0.8938	92.56	0.3922	0.9255
ResNet + BB + AB	90.00	0.3516	0.8979	95.33	0.2213	0.9534
ResNet + BB(DLDLM)+ AB(DLDLM)	90.00	0.3632	0.9002	94.44	0.2357	0.9444
ResNet+ BB(DLDLM) + AB	**90.67**	**0.3456**	**0.9058**	**96.78**	**0.1882**	**0.9677**

**Table 9 sensors-23-09390-t009:** Experimental results of different deep methods at each level with the RFID dataset.

Method	1 (0–50 m)	2 (50–100 m)	3 (100–200 m)	4 (200–500 m)	5 (500–1000 m)	6 (1000 m+)
AlexNet	84.00	88.00	70.00	78.00	**96.00**	98.67
VGG16	78.00	84.67	76.67	83.33	93.33	99.33
ResNet18	85.33	87.33	80.67	84.00	**96.00**	99.33
ResNet50	**86.67**	85.33	80.67	88.67	95.33	**100.00**
SCNN	85.33	85.33	**82.67**	82.67	95.33	98.67
TVRNet	82.67	84.67	78.67	88.00	94.67	98.67
VisNet	82.00	82.67	76.67	83.33	94.67	99.33
Ours	84.67	**89.33**	82.00	**92.00**	**96.00**	99.33

**Table 10 sensors-23-09390-t010:** Experimental results of different deep learning methods at each level with the FRIDA dataset.

Method	1	2	3	4	5	6	7	8	9
AlexNet	98.00	99.00	95.00	89.00	78.00	81.00	86.00	61.00	78.00
VGG16	98.00	98.00	92.00	85.00	78.00	79.00	74.00	65.00	71.00
ResNet18	98.00	95.00	98.00	90.00	94.00	88.00	85.00	76.00	84.00
ResNet50	**100.00**	92.00	95.00	91.00	91.00	87.00	91.00	79.00	78.00
SCNN	**100.00**	99.00	96.00	97.00	93.00	78.00	85.00	68.00	80.00
TVRNet	**100.00**	98.00	98.00	91.00	89.00	84.00	73.00	58.00	90.00
VisNet	**100.00**	**100.00**	**99.00**	97.00	91.00	82.00	82.00	75.00	76.00
Ours	**100.00**	**100.00**	**99.00**	**98.00**	**96.00**	**95.00**	**98.00**	**90.00**	**95.00**

**Table 11 sensors-23-09390-t011:** The run time per epoch and complexities for the proposed and baseline methods with the RFID dataset.

Method	Run Time (s/Epoch)		Time Complexities (GMacs)	Space Complexities (MB)	Parameters (M)
	Train Time (s/Epoch)	Validation Time (s/Epoch)			
SCNN	6.39	2.69	0.297	169	44.34
AlexNet	7.06	2.82	0.711	217	57.03
TVRNet	7.13	3.15	0.111	5.52	1.45
ResNet18	8.43	2.92	1.82	42.7	11.18
Ours	12.91	2.64	1.82	42.7	11.19
ResNet50	14.2	3.27	4.12	90	23.52
VGG16	19.22	3.69	15.5	512	134.29
VisNet	33.51	8.95	12.75	64.2	16.05

## Data Availability

The data used to support the findings of this study are available from the corresponding author upon request.

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
