# Peer review of "Visibility Estimation Based on Weakly Supervised Learning under Discrete Label Distribution"

_sensors, 2023, doi:10.3390/s23239390_

Round 1
Reviewer 1 Report
Comments and Suggestions for Authors
The authors propose a novel method for visibility estimation from foggy images based on weakly supervised learning and discrete label distribution learning. The key ideas are using bilinear attention pooling for weakly supervised localization to extract the farthest visible regions, and discrete label distribution learning to handle the ambiguity between adjacent visibility levels. The method is evaluated on two datasets - a real highway fog image dataset and a synthetic fog image dataset, showing superior performance over baseline methods like ResNet, VGGNet, and other visibility estimation networks. The paper is well-written and technically sound overall. I have some suggestions to further improve the paper:
1. The proposed weakly supervised localization module based on bilinear attention pooling is interesting, but lacks some details on implementation and analysis. First, how exactly is the attention map selected from multiple channels - is it just random? Some analysis on the spatial distribution of attention maps and which tends to focus on farther regions would help justify the approach. Second, the threshold for generating the visibility mask could be analyzed more thoroughly - how sensitive is performance to this parameter?
2. For the discrete label distribution learning, the authors use a Gaussian to model distribution over adjacent visibility levels. This assumes a symmetric spread of visibility, but the confusion matrix in Fig. 9 shows asymmetric confusion patterns. Have the authors experimented with more flexible distributions like mixture models to better capture asymmetry?
3. The results demonstrate clear improvements over baseline classifiers, but how does the method compare to other recent weakly supervised or few-shot visibility estimation methods, like [1,2]? Comparison with related methods would better highlight advantages of the proposed approach.
4. While accuracy and RMSE are standard metrics for visibility estimation, perceptual quality metrics like LPIPS could reveal if attention helps focus on perceptually relevant regions. Qualitative examples highlighting differences would also be insightful.
5. The complexity analysis in Section 4.4.5 is useful but quite brief - are there any tradeoffs in accuracy vs efficiency? How does varying number of attention maps or other parameters impact efficiency and accuracy?
6. Discussion could elaborate more on challenges and limitations, such as amount of training data needed, sensitivity to hyperparameters, types of scenes where the method succeeds or struggles.
7. Some references that could further enrich the related work: Robust Feature Matching for Remote Sensing Image Registration via Guided Hyperplane Fitting,Deterministic model fitting by local-neighbor preservation and global-residual optimization.
Overall the paper makes a nice contribution in advancing visibility estimation through integration of weakly supervised localization and label distribution learning. Addressing the above suggestions can further improve the analysis and highlight the advantages of the method.
Comments on the Quality of English LanguageSee above
Reviewer 2 Report
Comments and Suggestions for Authors
1). Section 2.2.: How the Bilinear Attention Pooling allows for better extraction of local features. It seems to me as a conjecture. The authors must justify the benefits of using the above pooling structure.
2). Section 2.2: The attention maps and the feature maps create a hybrid scheme called attention feature maps. Are there any possible correlations resulting from the above procedure? In case there are, the authors must discuss the way they handle these correlations.
3). The paper must include a rigorous comparative statistical analysis.
Comments on the Quality of English LanguageModerate English corrections are needed.
Round 2
Reviewer 1 Report
Comments and Suggestions for Authors
We think the authors have carefully considered my comments. This version is find for me.
Reviewer 2 Report
Comments and Suggestions for Authors
The authors answered all of my comments.
Comments on the Quality of English Language
Minor corrections